# Development of Model to Predict Natural Disaster-Induced Financial Losses for Construction Projects Using Deep Learning Techniques

**Ji-Myong Kim [1], Junseo Bae [2], Seunghyun Son [3], Kiyoung Son [4] and Sang-Guk Yum [5,***

[1] Department of Architectural Engineering, Mokpo National University, Mokpo 58554, Korea; jimy6180@gmail.com

[2] School of Computing, Engineering and Physical Sciences, University of the West of Scotland, Paisley PA1 2BE, UK; junseo.bae@uws.ac.uk

[3] Department of Architectural Engineering, Kyung Hee University, Suwon 17104, Korea; seunghyun@khu.ac.kr

[4] School of Architectural Engineering, University of Ulsan, Ulsan 44610, Korea; sky9852111@ulsan.ac.kr

[5] Department of Civil Engineering, Gangneung-Wonju National University, Gangneung 25457, Korea

* Correspondence: skyeom0401@gwnu.ac.kr

**Abstract:** This study goals to develop a model for predicting financial loss at construction sites using a deep learning algorithm to reduce and prevent the risk of financial loss at construction sites. Lately, as the construction of high-rise buildings and complex buildings increases and the scale of construction sites surges, the severity and frequency of accidents occurring at construction sites are swelling, and financial losses are also snowballing. Singularly, as natural disasters rise and construction projects in urban areas increase, the risk of financial loss for construction sites is mounting. Thus, a financial loss prediction model is desired to mitigate and manage the risk of such financial loss for maintainable and effective construction project management. This study reflects the financial loss incurred at the actual construction sites by collecting claim payout data from a major South Korean insurance company. A deep learning algorithm was presented in order to develop an objective and scientific prediction model. The results and framework of this study provide critical guidance on financial loss management necessary for sustainable and successful construction project management and can be used as a reference for various other construction project management studies.

**Keywords:** loss prediction model; construction project management; construction site; deep learning algorithm; deep neural network



## 1. Introduction

Recently, as the scale and complexity of construction works are increasing, variation of construction methods, and aggressive introduction of new construction methods are being made. As a result, various new risk factors for fiscal loss are occurring, and uncertainty in the financial risk prediction is increasing rapidly [1]. Therefore, the requirement for more dependable and scientific financial risk management in the complete construction project process is constantly being highlighted. However, current construction project risk management techniques are not adequately responding to these demands [2]. Most construction project risk assessments rely on individual and qualitative assessment grounded on views and knowledge of the orderer, individual contractor, and construction manager rather than on scientific analysis and substantiation [2,3].

In particular, on construction sites, various types of accidents occur and are exposed to the outside, and they are greatly affected by geographical and environmental factors, resulting in large and small personal injuries and physical losses [4]. For example, construction work near coastal, mountainous, and river areas, which is increasing with high preference, is greatly affected by geographic requirements. High-rise construction in a metropolitan city may inflict third-party losses on nearby buildings or pedestrians due to vibration and

noise generated at the construction site, falling objects, and flying objects. In addition, the increase in construction scale is causing an upsurge in fall accidents due to work at high places and safety accidents due to the use of heavy equipment. These factors are causing enormous economic losses despite the rapid development of new technologies and construction technologies of the fourth revolution such as Internet of Things, unmanned transportation, robot engineering, and 3D printing [5].

Nevertheless, the risk assessment and management of construction sites relies on subjective judgment or experience, and despite government and private sector policies, promotions, and investments to lower the high industrial accident rate in the construction industry, the construction industry remains the most dangerous industrial group [1,6,7]. Therefore, in order to prepare in advance for possible accidents and losses in construction sites, reduce risks, and establish strategies for transferring specific financial loss risks, analysis and prediction through scientific and empirical research should be carried out. In order to estimate the loss amount and loss range through this analysis and prediction, various and comprehensive risk factors are identified in advance throughout the entire construction process and quantified data that can be rationally collected is required [8–11].

Consequently, the purpose of this study is to develop a model that scientifically predicts the financial loss of construction sites based on more objective data. In other words, this study is to develop a model for predicting financial loss of a construction site by using a deep learning algorithm based on actual loss data generated at construction sites. These models and their framework are expected to contribute to the sustainable risk management of construction projects in the future.

## 2. Literature Review

The risk analysis of construction works is to create a strategy for proficiently investing limited financial resources of construction projects by managing potential risks in the construction site through prevention and reduction in advance [12]. In spite of this importance, risk analysis has a lower weight of research compared to topics such as cost management, quality management, and schedule management, which are the management elements of a construction project. Moreover, in research methods, qualitative methodology has conventionally been centered on qualitative methodology rather than quantitative methodology. Considering that the ultimate goal of risk analysis is the efficient allocation and input of defined resources, quantitative methodology is more suitable for risk analysis. In addition, the quantitative methodology enables a more realistic and accurate analysis of the potential risks of construction sites and enables higher generalization [13]. Furthermore, through the introduction of a quantitative risk analysis methodology, potential risk factors can be identified, factors indexed, and quantified models can be developed. For this reason, many risk analysis studies have begun to adopt a quantitative methodology. The construction industry has traditionally been classified as a high-risk industry. The reason is due to the specificity and complexity of construction works. In addition, there is also a synchronicity because several construction participants have to complete work within a fixed period [6]. For this reason, the risk analysis of construction works has a lot of uncertainty [14]. Thus, qualitative methods are inadequate for sophisticated predictions of risk, including risk uncertainty. Therefore, in order to reduce this uncertainty, quantitative data for a statistical and scientific approach and risk assessment and prediction through quantitative analysis methods are required [15]. However, qualitative research methods related to risk assessment in many construction projects are mainly used [16]. For example, Dikmen et al. identified risk factors using subjective assessment factors such as expert knowledge and experience in a construction risk assessment study [3]. Wood and Ellis also conducted risk assessments based on expert experience and judgment through surveys and checklists [17]. Baker et al. found that the opinions and experiences of experts, orderers, and engineers in the study of construction risk assessment techniques are the most frequently used techniques for assessment [18]. Warszawski and Sacks [19] found that sensitivity analysis is often used in construction project risk analysis due to the one-off

of construction projects and lack of data for analysis. They also argued that more advanced and efficient analysis methods are required. However, in many risk analysis studies, due to the lack of data and limitations in data collection, most risk assessments are performed using risk scales [20–23]. This demonstrates that the accuracy and reliability of the risk assessment can be increased when the actual loss amount is used in the risk assessment. As described above, past studies have suggested several categories of methods to risk assessment at construction sites, but subjective assessments based on personal judgment or expert opinions are the main focus. Therefore, quantitative evaluation through quantitative and reliable data is necessary for objective and scientific evaluation of construction site risk. Consequently, this study collected and analyzed the amount of insurance claim payouts that occurred at the actual construction sites of an insurance company. The insurance claim payouts are highly dependable. The reason is that the payment procedure is standardized and the amount is calculated through objective analysis by a certified loss assessor [5]. Furthermore, in this study, deep learning algorithms were used for objective and scientific analysis of data and model development, resulting in quantitative and reliable results.

Furthermore, it is difficult to predict the occurrence and extent of damage of a natural disaster. To cope with unexpected extreme events, several companies and nations have developed risk assessment tools. For example, the New Multi-Hazards and Multi-Risk Assessment Method for Europe (MATRIX) and the Probabilistic Risk Assessment have been developed for use in South America to mitigate damages from natural disasters. The United States is also leading nation in developing risk assessment models such as HAZUS-MH for multi-hazard risk assessment developed by the Federal Emergency Management Agency (FEMA) [24]. Florida, a hurricane-prone area in the United States, has developed Florida Public Hurricane Loss Prediction Model (FPHLM) to assess damages caused by hurricanes and to predict financial losses and casualties [25].

These tools can be utilized to assess potential damages to buildings and other infrastructures when extreme events such as earthquakes, floods, and so on occur. Moreover, Geographic Information System (GIS) can be supplemented to these risk assessment tools for more reliable data regarding estimating social and spatial damages caused by natural hazards. Various factors such as geospatial data, demographics, and revenues have been included in these models for evaluating possible extents of damages [26].

Many studies have been conducted to find valuable factors for estimating damages caused by natural hazards. Yum et al. [13] have proposed a methodology to find the most critical factors affecting tunnel construction projects. They utilized various atmosphere factors such as wind speed, rainfall, and so on to compare the actual amount of loss provided by insurance companies. Moreover, damage ratio for calculating the actual amount of damage and claim payout has been utilized to estimate maintenance and repair cost for accommodation facilities. One study has determined the maintenance cost of an international hotel chain caused by natural hazards such as floods, hurricanes, power outage, and so on using multiple regression analysis and damage ratio to determine the most influential risk factors for future repair costs [27].

Risk in construction projects cannot be determined by a single factor [28]. Therefore, many possible factors should be considered when estimating accurate damages such as financial losses and schedule delays. According to Hastak and Baim [29], the quality of workers meaning the extent of their training for specific construction projects is the main risk factor affecting construction projects. Safety and built environments are also key risk factors affecting construction projects. One study has used multiple linear regression method and neural network analysis to find the most influential risk factors [30].

Hashemi et al. [31] have found that failed project management is due to schedule delays, relationship issues among stakeholders, labor, and materials issues, and so on. Supporting findings of Hasemi wet al. [31], Li et al. [32] have proposed an economic building technology and reported that dispute issue is a critical construction risk factor. Many studies have been conducted regarding risk assessment for construction projects. However, most risk management tools have focused on developed countries where enough loss data

are available. More universe advance risk assessment method is required to be utilized in developing countries where data are insufficient. Moreover, unexpected events such as natural hazards should be considered simultaneously with project management factors since these two (i.e., natural hazards and project management) could be complementary factors to mitigate the potential risk at construction sites.

Artificial intelligence, unmanned transportation, big data, robotics, IoT, etc., which are emerging recently, have been applied in various fields and have been recognized for their effects [33,34]. The introduction of a new paradigm is indispensable for the improvement of the construction industry, which is classified as a dangerous industry group, and for the reduction of financial losses. Moreover, the demand for deep learning technology for analyzing vast amounts of big data generated from sensor information and various IoT devices that are widely used in recent construction sites is expected to increase exponentially. Therefore, in this study, for the introduction of a new paradigm for risk assessment in the construction industry and the development of deep learning technology for big data related to construction, a prediction model was developed using deep learning algorithms. This prediction model will contribute to improving the accuracy of prediction of losses occurring in future construction sites.

## 3. Research Goals and Methodology

The aim of this study is to generate a model that predicts the financial loss of a construction site using a deep learning algorithm founded on the financial loss data occurred at the construction site. As shown in Figure 1, the detailed aim is (1) to gather data on financial loss incurred at actual construction sites. (2) Based on the collected data, a loss prediction model is generated through a deep learning algorithm. (3) Validate the model through comparison with other model results. In the model verification, the results of the deep learning algorithm model and the results of commonly used a statistical model were compared. The detailed steps of the study are as follows: First, input variable and an output variable related to the cost of financial loss at the construction site are collected. Second, a deep learning algorithm model and a statistical model were established, individually. The two models calculated MAE (mean absolute error) and RMSE (square mean square error) values, respectively, and compared the two results. The deep learning model utilized Python 3.7, and the statistical model was generated using IBM Statistical Package for the Social Sciences (SPSS) V23.

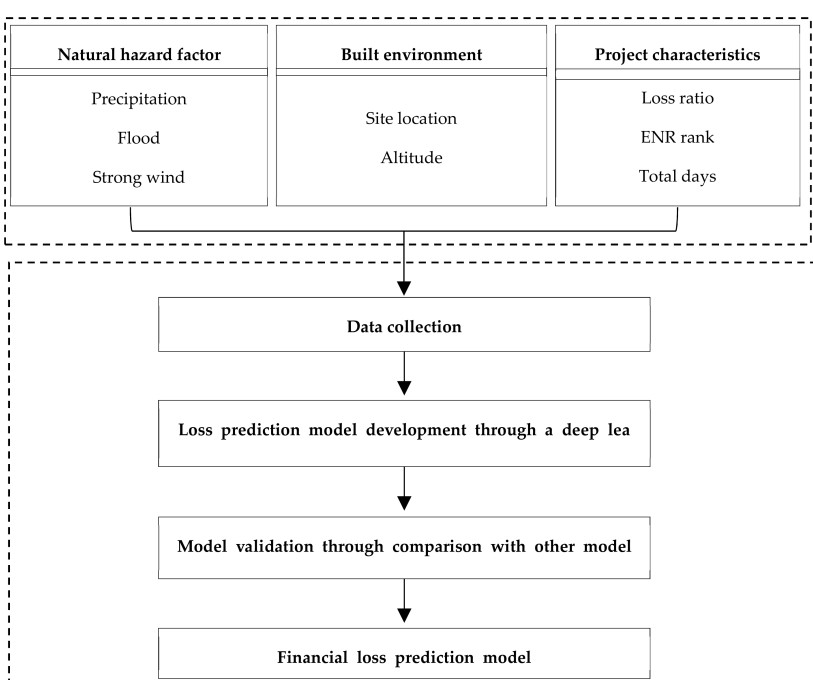

**Figure 1.** General approach and work flow of a prediction model.

## 4. Data Collection and Input Variables

This study accepts a record of claim payouts from an insurance company's Contractors' all-risks insurance (CAR). CAR is intended to comprehensively cover losses incurred at all stages of a construction project for contractors. The construction targets for CAR application are broad, including buildings, roads, ports, railways, bridges, tunnels, and plants, and include all periods related to construction from the start of construction to the commissioning period after construction is completed. The scope of application includes the construction of new construction and additional buildings, and it compensates for the contractor's economic losses incurred by the object of construction, materials and equipment, life, and third parties. The collected claim payout records target financial losses incurred at construction sites from 1999 to 2018, and the total number was 1930. The collected information does not include personal information. Only pure loss amount was used to exclude the difference in the amount of loss according to the insurance conditions of each individual subscriber. The record of claim payouts received from the insurance company included information on the amount of loss, details of the loss, the contractor, the date of the accident, the construction period, the construction amount, and the location. The scope of this study is restricted to South Korea.

Based on the collected data, information on loss amount and basic construction-related information (Engineering News Record rank, total days, progress rate, and total construction cost) and natural disaster risk (site location, altitude, precipitation, flood, and strong wind) was gathered. The natural disaster risk at the site was produced based on the location information of the collected data. A description of each indicator is shown in Table 1. The dependent variable is the loss ratio obtained by dividing the loss incurred in the construction project by the total construction cost. The Engineering News Record (ENR) Rank, total days, progress rate, total construction cost, and altitude were entered as numeric variables, correspondingly.

**Table 1.** Description of variables.

| Variables | Explanation | Unit |
|---|---|---|
| Loss ratio | Value obtained by dividing the amount of loss incurred in construction projects by each total construction cost | Number |
| ENR rank | Engineering News Record rank | Number |
| Total days | Total number of construction days | Number |
| Progress rate | Construction progress rate in case of loss | Number |
| Total construction cost | Total amount of construction cost (million KRW) | Number |
| Site location | Classification of the location of the construction site | Nominal 1.Suburban 2.Urban 3.Metropolitan |
| Altitude | Altitude above sea level (m) | Number |
| Precipitation | Risk rank of amount of precipitation at the site | Nominal - Rank 1-5: 1-5 |
| Flood | Risk rank of flood ground on the anticipated annually occurrence of flood at the site | Nominal - Rank 1-6: 1-6 |
| Strong wind | Risk rank of strong wind for maximum wind speed at the site (100-year return period) | Nominal - Rank 1-5: 1-5 |

The location was a nominal variables which is divided into three groups as suburban, urban, and metropolitan. The precipitation, flood, and strong wind were entered as nominal variables, individually, utilizing natural disaster risk ratings. Natural disaster risk ratings were evaluated using the risk levels for each natural disaster in Munich Reinsurance Company's Natural Risk Assessment Network (NATHAN) for an impartial

and dependable assessment. NATHAN is an online natural disaster risk map system created to estimate the risk of various natural disasters around the world. Through this system, it is possible to check the scientific and objective level of various natural disaster risk levels based on location information. Natural disaster risk level is scientific and has high reliability because it is the result of comprehensively utilizing the past severity and frequency of natural disasters, data from public organizations or institutions, and analysis results using analysis tools [CURIE 23]. The risk of each natural disaster was used as a nominal variable. The loss ratio, total construction cost was converted to natural logarithm for normal distribution of the data. Descriptive statistics of each variable are revealed in Table 2.

**Table 2.** Descriptive statistics of variables.

| Variables | N | Minimum | Maximum | Mean | Std. Deviation |
|---|---|---|---|---|---|
| Loss ratio | 1930 | −12.72 | 3.02 | −7.28 | 1.99 |
| ENR rank | 1930 | 1.00 | 100.00 | 37.97 | 41.14 |
| Total Days | 1930 | 121.00 | 4749.00 | 1338.93 | 889.76 |
| Progress rate | 1930 | 0.00 | 20.44 | 0.02 | 0.52 |
| Total construction cost | 1930 | 0.69 | 18.20 | 11.19 | 1.75 |
| Site location | 1930 | 1.00 | 3.00 | 1.88 | 0.84 |
| Altitude | 1930 | −3.00 | 910.00 | 84.25 | 148.62 |
| Precipitation | 1930 | 2.00 | 5.00 | 4.63 | 0.69 |
| Flood | 1930 | 0.00 | 5.00 | 1.75 | 1.71 |
| Strong wind | 1930 | 1.00 | 5.00 | 1.37 | 0.53 |

## 5. Deep-Learning Algorithm Model

Deep learning is a technology that implements type classification or regression through machine learning of input data and is broadly accepted in forecast and recognition areas. The deep learning model is composed of the input layer, output layer, and hidden layer, activation function, weight, and neuron, and can be applied to various data because it can have a neural network composed of various structures and layers [35,36]. Deep learning is classified according to its structure and processing method, and representatively, there are Generative Adversarial Network (GAN), Convolutional Neural Network (CNN), Auto Encoder (AE), Deep Neural Network (DNN), and Recurrent Neural Network (RNN). For example, DNN is a typical neural network with varied numbers of hidden layers, which trains to model compound nonlinear interactions [37,38]. Owing to the features, DNNs can be demonstrated in various types of artificial neural networks and are extensively used in forecast and labeling in various businesses and educational fields [39]. This study developed a model for predicting financial loss of construction sites using DNN in respect of input data, output format, and universality of the model.

The developed model was verified by calculating RMSE (root mean square error) and MAE (mean absolute error). The RMSE and MAE values represent the errors between the predicted and actual results and are key indicators for evaluating the artificial neural network model [40]. MAE is an absolute value estimated and averaged the alteration between the predicted value and the actual value. The smaller the MAE value, the smaller the prediction error. RMSE is expressed as a solitary measure of the variance between the predicted and actual values of the model, and the smaller the RMSE value, the smaller the prediction error. The input data was preprocessed using the z-score normalization method in order to control different units and quantities between variables. Of the total data, 70% was used as learning data (30% of the data was used as validation data), and 30% was utilized as test data.

### 5.1. Network Structure Scenario and Hyper-Parameter Tuning

Since the DNN model adjusts the model using a backpropagation algorithm, the optimal combination may differ depending on the input and output variables. In order to find the optimal model, it is necessary to find the optimal model arrangement through trial and error through network structure scenario and hyper-parameter tuning [41]. The network architecture scenario will determine the number of layers and nodes. The hyper parameters decide the dropout, optimizer, batch size, epoch, and activation function. For example, dropout is a normalization penalty to prevent overfitting, which degrades the performance of deep learning models. The optimizer is concerned with the speed and stability of learning, and the batch defines the unit of learning for efficient learning. Epoch specifies the number of learning. The activation function specifies how to adjust the weights of the nodes to find the least cost function [41,42].

In this study, considering the amount of data, the network structure scenario is set to have three hidden layers, and the dropout is determined to be 0 or 0.2. The optimal combination was simulated with trial and error. The batch was designated 5 and the epoch was assigned 1000. Adaptive Moment Estimation (Adam) was used as the optimizer, the ReLu (Rectified Linear Unit) function was accepted as the activation function. Adam Method is a generally widely used optimization algorithm due to its efficiency of calculation and versatility. It is a first-order gradient algorithm with the concept of a moment in a stochastic objective function [43]. ReLu was developed to solve the shortcomings of the existing Sigmoid function as an activation function that changes the output according to the input value greater than or equal to zero [44].

Table 3 represents the MAE and RMSE value for the learning results for each network structure scenario and dropout (0, 0.2). A scenario in which MAE and RMSE have minimum values was selected as the final model. As a result of learning, when the dropout value is 0.2, the loss function is commonly greater than when the dropout value is zero. In addition, as the number of hidden layer nodes growths, the MAE and RMSE values incline to growth, and in the scenario where the number of hidden layer nodes is 700-700-700, both MAE and RMSE have minimum values, and the MAE and RMSE values have a tendency to increase again. Consequently, the Network Structure Scenario of the final model was determined to be 700-700-700, and the dropout was zero.

**Table 3.** Result of learning.

| Network Structure Scenario | Dropout (0) | | Dropout (0.2) | |
|---|---|---|---|---|
| | MAE | RMSE | MAE | RMSE |
| 100-100-100 | 1.016 | 1.281 | 1.442 | 1.818 |
| 200-200-200 | 0.935 | 1.175 | 1.476 | 1.874 |
| 300-300-300 | 1.009 | 1.308 | 1.439 | 1.789 |
| 400-400-400 | 0.986 | 1.237 | 1.479 | 1.864 |
| 500-500-500 | 0.924 | 1.159 | 1.355 | 1.698 |
| 600-600-600 | 0.918 | 1.222 | 1.417 | 1.796 |
| 700-700-700 | 0.855 | 1.072 | 1.341 | 1.684 |
| 800-800-800 | 0.911 | 1.133 | 1.373 | 1.772 |
| 900-900-900 | 0.897 | 1.131 | 1.299 | 1.657 |
| 1000-1000-1000 | 0.978 | 1.253 | 1.282 | 1.620 |

### 5.2. Final Model and Validation

Table 4 shows the firmed network structure and hyper parameters. The MAE and RMSE values were calculated and compared, individually, using verification data and test data for the validation of the DNN model. In addition, the Multiple Regression Analysis (MRA) model was established utilizing the Multiple Regression Analysis Method, which is commonly adopted in the generation of the current prediction model. MAE and RMSE values were estimated, separately, and the values of the two models were matched. The IBM Statistical Package for the Social Sciences (SPSS) V23 was accepted to create for MRA

model. The results and comparison results of each model as shown in Table 5. The results of verification data were MAE 0.707 and RMSE 0.844, and the results of test data were MAE 0.774 and RMSE 0.975 in the DNN model. As a result of comparing the values of the two data, the overfitting problem of the model is considered to be insignificant since the difference concerning the two values is not great. Furthermore, the DNN model had a minor prediction error rate of 11.2% in MAE and 42.2% in RMSE than in MRA as a result of comparing the two models.

**Table 4.** Configuration of network structure and hyper-parameter.

| Group | Composition | Detail |
|---|---|---|
| Network structure | Layer | 3 |
| | Node | 700-700-700 |
| Hyper Parameter | Optimizer | Adaptive Moment Estimation Method |
| | Activation Function | Rectified Linear Unit function |
| | Dropout | 0.0 |
| | Batch Size | 10 |
| | Epoch | 1000 |

**Table 5.** Model comparison result.

| Model | Validation | | Test | |
|---|---|---|---|---|
| | MAE | RMSE | MAE | RMSE |
| DNN | 0.707 | 0.844 | 0.774 | 0.975 |
| MRA | - | - | 0.861 | 1.386 |
| DNN/MRA (%) | | | 11.2% | 42.2% |

## 6. Discussion

In this study, a model for predicting financial loss of construction sites was developed using the DNN algorithm, one of the deep learning algorithms. For model development, an insurance company's claim payout record was recorded to collect data on the cost of financial losses incurred at the actual construction site. A deep learning model was trained based on the collected data. Moreover, the proposed model was compared with other models to validate its effectiveness. Moreover, in order to derive an optimal model, a trial-and-error method was adopted to find the network scenario and hyper-parameters. As a result of model comparison, the DNN model was 11.2% minor in MAE and 42.2% in RMSE than the MRA model (0.861, 1.386), correspondingly. Consequently, it can be seen that the non-parametric model DNN is more suitable than the parametric model MRA for the analysis of financial loss data of construction sites with nonlinear characteristics. Furthermore, in the comparison of the ratio of the two indicators, RMSE showed a prediction error rate of 30% or more lower than that of MAE, which is due to the calculation characteristic of RMSE, i.e., giving a large penalty to a large error value. This result indicates that the non-parametric model DNN is more reliable than other models for identifying the extent of financial risk for construction projects. Owing to the nature of the construction industry, the size of the project varies widely, so this error value may occur more frequently. It can be seen that the learning of the DNN model reflects these singular values well.

Using the methodology and development model of this study, the manager of a construction site can predict the financial loss cost of their construction sit, or can develop an optimal deep learning prediction model according to the needs and conditions of managers at other construction sites. Moreover, in terms of the precision of prediction, the DNN model has a lower prediction error rate than the current model, so it will be more reliable and capable of precise cost prediction. Based on these sophisticated financial loss predictions, it is possible to reinforce the management of the construction site through

active investment to reduce the amount of loss by grasping the amount of risk in advance. For instance, it is possible to estimate the amount of financial loss before the start of construction, prepare countermeasures for losses, and establish preventive strategies for facility investments in advance. In addition, it will be helpful in the stages of financial planning such as project budget preparation and emergency reserves based on sophisticated loss cost forecasting. For case, project owners will set a guideline on risk that fits their risk appetite and capital assets and help manage business continuity. Through the established guideline, it is likely to prepare strategies for avoidance and transfer of losses, such as expanding insurance coverage and purchasing special contracts according to the expected loss. This could reduce the risk of possible financial losses in the future. Moreover, it will provide a standard for thinking about the rate level for the currently subscribed insurance rate or the insurance rate to be subscribed.

The established model is able to be useful to the financial loss cost analysis and prediction model development of other industrial sites in the future. Therefore, it can be used for the development of models or systems in the private or public sector of other industrial sites. Moreover, natural disasters such as earthquakes, hurricanes, and so on have been well recognized as game changers with negative effects on societies, economics, and environments at national levels recently due to the rapid climate change [45–48]. Previous studies have insisted that natural hazards have effects on construction projects and actual sites. For example, Kim et al. [49,50] have revealed that typhoon-induced heavy wind and distance from shorelines have close relationships with significant damages to residential and commercial buildings at the region where hurricanes occur. Moreover, movement and direction of hurricanes can have significant effects on the extent of damage for buildings. In this study, natural disasters such as flood, precipitation, and maximum wind speed have been considered as variables that can affect financial loss at the construction site additionally. The DNN model revealed that natural disasters could be associated with financial losses. This finding can be utilized for mitigating unexpected risks such as financial losses since natural disasters would be the strongest influential factors for losses at construction projects [13]. Future construction projects can utilize findings of this study to cope with various possible natural hazards for preventing unexpected financial losses while considering additional more specific construction site-oriented variables such as geographical and local weather. Although these factors could not be easily estimate quantitatively, if they could be incorporated into risk assessment models, future construction projects could be protected, thus preventing unexpected financial losses and casualties at actual construction sites.

In addition, it will be a basic study to analyze the vast amount of big data generated from various IoT devices, sensor devices, CCTVs, etc., which are widely spread in recent construction sites. However, this study adopted a record of claim payouts from an insurance company in South Korea. Further research is desired to collect, compare, and prove claim payout records from diverse countries or different insurance companies in the future. Additional research is preferred through the amount of data and additional input variables through data collection of additional loss data and additional variables for the advancement of the developed model. This research was not performed to find out the relationship between the main valuable and the type of construction projects due to inherent limitations of the model used in this study. If additional future studies could accumulate more loss data on various types of construction projects, it could be possible to match specific causes for financial losses with different kinds of construction projects and prepare well-customized plans to cope with unexpected situations associated with financial losses.

Furthermore, future research should consider both aleatory and epistemic uncertainties for random variables to have more reliable predictions for financial losses. For example, the epistemic uncertainty can occur due to the lack of available data or knowledge. However, such uncertainty can be reduced by applying high-quality data such as empirical data (i.e., insurance claim payout) regarding specific construction projects. Therefore, future

research that applies more reliable probabilistic methodologies and reliable data is strongly recommended to obtain robust results regarding prediction of financial losses.

## 7. Conclusions

In recent years, the scale of construction sites has increased, and high-rise buildings and complex building construction projects are increasing. In addition, the recent increase in natural disasters and urbanization caused by global warming pose a risk of construction projects. Therefore, both the severity and frequency of accidents occurring at construction sites are increasing, and accordingly, financial losses are increasing rapidly. Consequently, for effective and sustainable construction project management, a model for predicting financial losses to reduce and manage the risk of such financial losses is inevitable. In this study, a financial loss model of a construction site was developed using a deep learning algorithm based on the financial loss data of a construction site.

In this study, a model was developed by applying a deep learning algorithm to predict the financial loss cost of a construction site and verified through comparison of the results of other models. As a result of the validation, the model established in this study can increase the reliability of the prediction of the cost of financial loss at the construction site, and the prediction method can be improved. Hence, a model for predicting financial loss of a construction site utilizing deep learning technology can be a key to effective construction project management and reduction of the risk of financial loss. Since the results and framework of this study can be applied to other types of industrial sites and related studies, it will eventually help to reduce the cost of financial losses in the industrial sites. In addition, the model developed in this study can be advanced into a more reliable model through continuous data acquisition and additional verification with other models in the future.

Furthermore, the developed model in this study could be used for developing countries where historic loss data might be unavailable or insufficient to obtain reliable results for financial losses. Various factors such as natural hazards, built environments, insurance data, and project characteristics handled in this research provide a vital reference to government agencies to prepare guideline for coping with unexpected financial losses caused by various factors that can be found at construction sites. In particular, insurance companies could utilize results from this study such as applying risk variables that can affect their business plans in an area that is prone to natural hazards to reduce financial losses by setting various insurance techniques such as premium price and maximum loss, event limiting their asset under management (AUM). The insurer also can benefit from the premium price since they would have the opportunity to be paid for damages caused by unexpected natural hazards at construction sites. Such advantages can help both insurance companies and insurers to reduce their financial losses, reflecting risk variables revealed in this paper. Moreover, governments can apply results of the present study to prepare safety guideline and regulations to mitigate potential damages at construction sites. By doing so, they can enhance the resilience against potential numerous risks caused by the natural hazards.

Advanced risk assessment and management for construction sites could help the government to prepare unexpected extreme events such as natural hazards by providing more structured and reasonable risk mitigation plans for their whole communities. Moreover, developing countries where reliable data are unavailable may be able to utilize techniques and results presented in this study to prepare their own emergency and business plans to enhance safety and mitigate potential financial losses.

**Author Contributions:** Conceptualization, J.-M.K.; Data curation, S.-G.Y., S.S. and K.S.; Funding acquisition, J.-M.K.; Investigation, J.-M.K. and S.-G.Y.; Methodology, J.-M.K. and J.B.; Software, J.-M.K. and S.-G.Y.; Validation, S.-G.Y. and S.S., K.S.; Writing—original draft, J.-M.K.; Writing—review and editing, J.-M.K., S.-G.Y., S.S., K.S. and J.B. All authors have read and agreed to the published version of the manuscript.

**Funding:** This research was funded by Basic Science Research Program through the National Research Foundation of Korea (NRF) funded by the Ministry of Education (NRF-2019R1F1A1058800).

**Institutional Review Board Statement:** Not applicable.

**Informed Consent Statement:** Not applicable.

**Data Availability Statement:** Not applicable.

**Conflicts of Interest:** The authors declare no conflict of interest.

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
