# Peer review of "Development of Model to Predict Natural Disaster-Induced Financial Losses for Construction Projects Using Deep Learning Techniques"

_sustainability, doi:10.3390/su13095304_

Round 1
Reviewer 1 Report
In this paper, the authors develop a model for predicting financial loss at construction sites using a deep learning algorithm. The data for analysis are collected from claim payout data from a major South Korean insurance company, being a total of 1930. It is an interesting and current topic in the construction sector.
It is recommended to include a graphic to reinforce the explanation of the methodology.
In the discussion section, the explanation of the results and how they can provide an advantage for future projects could be improved. What variables are those that allow a new project to be compared with previous ones in order to predict financial losses?
In general, the analysis sounds very interesting and can have great potential.
Reviewer 2 Report
The article concerns the financial risk assessment in construction projects. A quantitative approach to this assessment has been proposed. COMMENTS: 1. Financial losses are limited only to the risk factors related to natural disasters on the construction site, if so, the title should be clarified. 2. The literature review on tools for quantifying the risk of construction projects has not been completed, although there are already some such tools. 3. What about other risk factors that are not subject to the so-called "law of large numbers" and whose probability and effects can rather be assessed subjectively for a specific project. Therefore, a quantitative approach to risk assessment in construction projects is problematic. 4. It would be useful to present the proposed tool on a specific example of a construction project and, based on the information received, to propose measures to prevent the occurrence of the forecast financial losses in this specific situation. The presentation of the use of the developed approach is very important from a practical point of view
Reviewer 3 Report
The paper presents the development of a model for predicting financial losses on construction sites using the DNN algorithm. The research is based on a wide research sample of 1930 records on the payment of insurance claims for damages incurred on construction sites in the years 1999 to 2018. Within the development of the model were identified basic variables (loss ratio, ENR rank, total days, progress rate, total construction costs, site location, altitude, precipitation, flood and strong wind) and described their basic statistical parameters (min., max., mean and std. deviation). The model 700-700-700 of the number of nodes of the hidden layer was identified as the optimal network structure scenario, which resulted from the data of Table 3 Result of learning.
The structure of the article is fine, abstract corresponds to the text, keywords are meaningful. The paper is strongly focused on the description of the model from statistical and mathematical perspective (well prepared and commented), but the reader somewhat lacks a description of the links between the variables and especially a specific technical verification study, or an example of determining of financial losses for particular construction. This should be added to the paper.
Round 2
Reviewer 1 Report
All the suggestions proposed in the first version have been carried out correctly.